# Mendelian Randomization Study of Lipid Metabolites Reveals Causal Associations with Heel Bone Mineral Density

**DOI:** 10.3390/nu15194160

**Published:** 2023-09-27

**Authors:** Mingxin Wu, Yufei Du, Chi Zhang, Zhen Li, Qingyang Li, Enlin Qi, Wendong Ruan, Shiqing Feng, Hengxing Zhou

**Affiliations:** 1National Spinal Cord Injury International Cooperation Base, Tianjin Key Laboratory of Spine and Spinal Cord Injury, Department of Orthopedics, Tianjin Medical University General Hospital, Tianjin 300070, China; 2Department of Endocrinology and Metabolism, Tianjin Medical University General Hospital, Tianjin 300070, China; 3Department of Orthopaedics, Qilu Hospital of Shandong University, Shandong University Centre for Orthopaedics, Advanced Medical Research Institute, Cheeloo College of Medicine, Shandong University, Jinan 250013, China

**Keywords:** bone mineral density, lipid metabolism, mendelian randomization, instrumental variables, single nucleotide polymorphism

## Abstract

Background: Osteoporosis, which is a bone disease, is characterized by low bone mineral density and an increased risk of fractures. The heel bone mineral density is often used as a representative measure of overall bone mineral density. Lipid metabolism, which includes processes such as fatty acid metabolism, glycerol metabolism, inositol metabolism, bile acid metabolism, carnitine metabolism, ketone body metabolism, sterol and steroid metabolism, etc., may have an impact on changes in bone mineral density. While some studies have reported correlations between lipid metabolism and heel bone mineral density, the overall causal relationship between metabolites and heel bone mineral density remains unclear. Objective: to investigate the causal relationship between lipid metabolites and heel bone mineral density using two-sample Mendelian randomization analysis. Methods: Summary-level data from large-scale genome-wide association studies were extracted to identify genetic variants linked to lipid metabolite levels. These genetic variants were subsequently employed as instrumental variables in Mendelian randomization analysis to estimate the causal effects of each lipid metabolite on heel bone mineral density. Furthermore, metabolites that could potentially be influenced by causal relationships with bone mineral density were extracted from the KEGG and WikiPathways databases. The causal associations between these downstream metabolites and heel bone mineral density were then examined. Lastly, a sensitivity analysis was conducted to evaluate the robustness of the results and address potential sources of bias. Results: A total of 130 lipid metabolites were analyzed, and it was found that acetylcarnitine, propionylcarnitine, hexadecanedioate, tetradecanedioate, myo-inositol, 1-arachidonoylglycerophosphorine, 1-linoleoylglycerophoethanolamine, and epiandrosterone sulfate had a causal relationship with heel bone mineral density (*p* < 0.05). Furthermore, our findings also indicate an absence of causal association between the downstream metabolites associated with the aforementioned metabolites identified in the KEGG and WikiPathways databases and heel bone mineral density. Conclusion: This work supports the hypothesis that lipid metabolites have an impact on bone health through demonstrating a causal relationship between specific lipid metabolites and heel bone mineral density. This study has significant implications for the development of new strategies to osteoporosis prevention and treatment.

## 1. Introduction

Osteoporosis, as a worldwide public health issue, is characterized by a decrease in bone mineral density (BMD), resulting in weakened bone strength and an increased risk of brittle fractures [1]. Osteoporotic fractures, particularly in the hip, spine, and wrist, are associated with higher morbidity, mortality, and reduced quality of life [2]. Current approaches for treating osteoporosis focus on reducing fracture risk and preserving bone health [3]. Early detection, prevention, and appropriate management are crucial in minimizing the impact of osteoporosis on individuals and society as a whole. BMD is a key predictor of osteoporotic fractures, with low BMD indicating a higher fracture risk [4,5,6]. Previous studies have shown that heel bone mineral density (H-BMD) serves as a useful surrogate for hip BMD [7]. Quantitative calcaneal ultrasound has been used to predict the risk of total and hip fractures in both males and females [8]. When spinal deformities make it difficult to measure BMD through quantitative computed tomography(CT) or other methods, the calcaneus can serve as an additional site for assessing bone density and evaluating bone loss in patients [9]. 

Lipid metabolites, including those involved in fatty acid metabolism, glycerol metabolism, inositol metabolism, bile acid metabolism, carnitine metabolism, ketone body metabolism, sterol and steroid metabolism, etc., have been hypothesized to affect bone metabolism based on numerous cross-sectional and mechanistic studies [10,11,12,13,14]. For example, some studies have shown a negative correlation between omega-6 fatty acids and total body BMD [10]. The risk of osteoporosis has also been linked to higher levels of total cholesterol and triglycerides [11]. Early postmenopausal women with an atherogenic lipid profile have been reported to have lower lumbar and femoral BMD and a higher risk of osteopenia than those with a normal lipid profile [12]. Significant differences in lipid metabolite content have been observed among individuals with different H-BMD groups [13]. Metabolic changes in the calcaneal bone marrow have also been found to be highly active in individuals with different osteoporosis states [14]. However, while these studies indicate a correlation between metabolism and heel or total BMD, the causal relationship between lipid metabolites and H-BMD remains unknown.

Mendelian randomization (MR), similar to a randomized controlled trial, is a statistical method that uses genetic variation to investigate the causal relationship between exposures and outcomes [15]. It utilizes single nucleotide polymorphisms (SNPs) associated with the exposures of interest as instrumental variables (IVs) to assess potential causal effects between exposures and outcomes, overcoming limitations of traditional observational studies [16]. MR has the advantage of temporal causality, meaning that the cause precedes the effects and is not influenced by confounding factors such as environmental exposures and behaviors. Two-sample MR analysis, using exposure-related IVs and outcome-related IVs from two different population datasets, can enhance statistical power [17]. Furthermore, metabolite-specific genome-wide association study (mGWAS) has revealed the genetic basis for metabolite features, enabling researchers to investigate the causal relationship between metabolites and illnesses or the key predictors of illnesses using MR analysis.

In this study, a two-sample MR analysis was performed to investigate the relationship between 130 lipid metabolites and H-BMD. In order to test the robustness of our results and take into consideration possible causes of bias such horizontal pleiotropy, we also conducted sensitivity studies. Among the 130 lipid metabolites analyzed, 8 showed a significant causal relationship with H-BMD. Among these, 1-arachidonoylglycerophosphocholine had a negative effect on H-BMD, but elevated levels of the remaining seven lipid metabolites had positive effects on H-BMD. We also investigated whether there is a causal relationship between the downstream metabolites in the pathway in which lipid metabolites are located and H-BMD. Interestingly, no causal relationship was found between downstream metabolites associated with a specific metabolite and H-BMD. This study may provide important insights into the causal relationship between lipid metabolites and H-BMD, ultimately offering theoretical references and new research directions for the prevention of osteoporosis.

## 2. Materials and Methods

### 2.1. Study Design and Data Source

We employed a two-sample MR method to examine the causal association between lipid metabolites and H-BMD among a study population of Europeans. In the MR analysis, we employed the inverse variance weighted (IVW) analysis to assess the relationships between IVs and outcomes. Additionally, we also performed sensitivity analyses using the MR-Egger and weighted median methods.

The data utilized in this study were sourced from publicly available datasets (https://gwas.mrcieu.ac.uk/, accessed on 31 August 2023) and obtained ethical approval in previous studies [18,19]. The exposure data were extracted from genome-wide association study (GWAS) data published by Shin et al. in 2014, which included 7824 Europeans [18]. By conducting genome-wide association studies of metabolites in blood samples, Shin and his colleagues give an extensive study of the influences of genes on human metabolism. Following strict quality control procedures, we selected 130 lipid metabolites for further investigation. The outcome data was based on the findings of a large-scale GWAS meta-analysis conducted by Morris et al. in 2019, which included 426,824 individuals from the UK Biobank and utilized around 13.7 million SNPs for association analysis [19]. In addition, quantitative ultrasound (QUS) of the heel was used to obtain the H-BMD data. The study was carried out following the guidelines of Strengthening the Reporting of Observational Studies in Epidemiology using Mendelian Randomization (STROBE-MR) [20].

### 2.2. Instrument Variables Selection

The SNPs related to lipid metabolites were obtained from a large-scale genome-wide meta-analysis of human blood metabolites [18]. The H-BMD cohort consisted of 426,824 European subjects. Initially, SNPs were selected from the pool of 130 lipid metabolites, and SNPs which showed significant associations with lipid metabolites at a genome-wide significance level (*p* < 5 × 10^–8^) were chosen. To ensure independence, a linkage disequilibrium analysis was performed with a threshold of r^2^ < 0.001. Moreover, in order to exclude the weak instruments, we also computed the F statistics for each IV and eliminated those with F-statistics < 10 [21]. The H-BMD datasets were subsequently utilized to extract the SNP-outcome effects. SNPs with mismatched alleles, palindromic SNPs, and SNPs with missing values were eliminated from the alignment of SNP-exposure and SNP-outcome effects, and any inverted SNP strands were oriented appropriately. The SNPs selected as IVs needed to satisfy three MR assumptions: (1) the genetic variation is related to risk factors; (2) genetic variation is not related to confounding factors; and (3) the genetic variant affects the outcomes only through the risk factors [16].

### 2.3. Metabolomic Pathway Analysis

The metabolic pathways associated with the 8 metabolites, which were found to have a causal relationship with H-BMD, were retrieved from the KEGG database (https://www.genome.jp/kegg/, accessed on 31 August 2023)) and WikiPathways database (https://www.wikipathways.org/, accessed on 31 August 2023). Information on other metabolite molecules that may be influenced by the downstream metabolites of the 8 identified metabolites was collected based on the upstream and downstream relationships in these metabolic pathways. SNP information corresponding to the downstream metabolites was obtained from the IEU GWAS database (https://gwas.mrcieu.ac.uk/about/, accessed on 31 August 2023) for conducting two-sample Mendelian randomization analysis.

### 2.4. Mendelian Randomization Analysis

A two-sample MR analysis was performed by using multiple methods including the IVW method [22], MR-Egger method [23], weighted median method [24], simple mode method [25], and weighted mode method [26] to assess the impact of lipid metabolites on H-BMD. The IVW method combines Wald estimates for each SNP using a meta-analysis approach to obtain an overall estimate of the effect of lipid metabolites on H-BMD [27]. To account for potential bias due to horizontal pleiotropic effects, we utilized the MR-Egger method and weighted median method to analyze and test for directional bias resulting from pleiotropy [24,28]. The MR-Egger method generates a weighted linear regression between the exposure coefficient and the outcome coefficient to better assess pleiotropy. The weighted median method provides a reliable estimation of the causal relationship when the weight in the analysis is equal to or greater than 50%. This method allows for some invalid genetic variations, as long as at least half of the instruments are valid [29]. Under the assumption that instrument strength is independent of direct effects (internal), the slope of the regression line represents an asymptotically unbiased causal estimation. Additionally, the intercept of the MR-Egger regression line can quantify and indicate horizontal pleiotropy in the overall genetic instrument data [24,30]. The intercept of the regression line being unequal to 0 can be used to detect horizontal pleiotropy using the MR-Egger method.

### 2.5. Analysis Software

The analyses were implemented using R (v 4.0.1), and the following R packages were utilized: TwoSampleMR (v0.5.7) [31], ggplot2 (v3.4.2), and forestplot (v3.1.1). A significance level of *p* < 0.05 was considered statistically significant.

## 3. Results

### 3.1. Causal Association Identified between Eight Lipid Metabolites and H-BMD

Among the 130 lipid metabolites we analyzed (including 124 known and 6 unknown) (Appendix A), we identified a total of 8 metabolites associations at the nominal significance level (IVW, *p* < 0.05) for H-BMD in the IVW model (Figure 1), including hexadecanedioate (OR_IVW_ = 1.171 [1.036, 1.324]), myo-inositol(OR_IVW_ = 2.057 [1.082, 3.909]), acetylcarnitine (OR_IVW_ = 1.169 [1.032, 1.325]), propionylcarnitine (OR_IVW_ = 1.239 [1.038, 1.477]), 1-linoleoylglycerophosphoethanolamine (OR_IVW_ = 1.347 [1.057, 1.716]), epiandrosterone sulfate (OR_IVW_ = 1.080 [1.014, 1.150]), 1-arachidonoylglycerophosphocholine (OR_IVW_ = 0.795 [0.658, 0.961]) and tetradecanedioate (OR_IVW_ = 1.128 [1.011, 1.259]). Among these, 1-arachidonoylglycerophosphocholine had a negative effect on H-BMD (OR_IVW_ = 0.795 [0.658, 0.961]). Elevated levels of the remaining seven lipid metabolites had positive effects on H-BMD.

### 3.2. Causal Relationship between Downstream Metabolites and H-BMD

To further elucidate whether these metabolites, which have a causal relationship with H-BMD, ultimately affect H-BMD through other intermediate factors, we utilized the KEGG and WikiPathways databases to identify downstream metabolites in the metabolic pathways that these metabolites may influence (Appendix A). Among them, hexadecanedioate, propionylcarnitine, and epiandrosterone sulfate were identified as terminal metabolites in their respective signaling pathways, while 1-arachidonoylglycerophosphocholine, tetradecanedioate, and 1-linoleoylglycerophosphoethanolamine were not found in any pathway within the KEGG and WikiPathways databases. Myo-inositol and acetylcarnitine, on the other hand, were identified as intermediate metabolites involved in multiple signaling pathways (myo-inositol: phosphatidylinositol signaling system (KEGG), glycerophospholipid biosynthetic pathway (Wikipathway) and inositol phosphate metabolism (KEGG); acetylcarnitine: alanine and aspartate metabolism (Wikipathway) and insulin resistance (Wikipathway)). Therefore, we investigated the casual relationship between these two metabolites, their downstream metabolites involved in the pathways, and H-BMD. Interestingly, none of these downstream metabolites showed a causal relationship with H-BMD (Figure 2).

### 3.3. The Robustness of MR Results: Sensitivity Analysis

We conducted sensitivity studies on these potential lipid metabolites to examine their heterogeneity and pleiotropy, aiming to assess the viability of MR. We assessed the heterogeneity of IVs using Cochran’s Q test, with the Q value representing the presence of heterogeneity. The results indicated that several IVs exhibited heterogeneity: hexadecanedioate (Q value = 0.003), myo-inositol (Q value = 0.001), acetylcarnitine (Q value = 0.104), propionylcarnitine (Q value = 0.021), 1-linoleoylglycerophosphoethanolamine (Q value = 0.001), epiandrosterone sulfate (Q value = 0.047), 1-arachidonoylglycerophosphocholine (Q value = 0.018), and tetradecanedioate (Q value = 0.0002). We initially attributed this heterogeneity to insufficient sample size and a limited number of SNPs for the IVs. Furthermore, to assess and correct for directional pleiotropy, we conducted MR-Egger regression analysis. The results indicated that, except for the IVs with a limited number of corresponding SNPs (myo-inositol, acetylcarnitine, propionylcarnitine, 1-linoleoylglycerophosphoethanolamine, epiandrosterone sulfate), no evidence of horizontal pleiotropy was found for the other IVs (1-linoleoylglycerophosphoethanolamine (Q value = 0.794), tetradecanedioate (Q value = 0.753), tetradecanedioate (Q value = 0.557)). This suggests that there is no influence from other confounding factors.

## 4. Discussion

Although several MR studies, such as those of Chen et al. [32], Wang et al. [10], and Tao et al. [33], have reported associations between metabolites and BMD (including central BMD, lumbar spine, and BMD), the causal relationship between lipid metabolites and osteoporosis or peripheral H-BMD remains unclear. Based on GWAS datasets from different cohorts, this work comprehensively investigates the causal association between 130 lipid metabolites and H-BMD. We also performed sensitivity analysis to assess the robustness of our results, particularly regarding potential sources of bias such as horizontal pleiotropy. Among all the exposure variables, eight showed a significant causal relationship with H-BMD. Acetylcarnitine, propionylcarnitine, hexadecanedioate, tetradecanedioate, myo-inositol, epiandrosterone sulfate, and 1-linoleoylglyceroptoethanolamine exhibited protective effects on H-BMD, while 1-arachidonoylglycerophospholine showed a negative correlation with H-BMD. On one hand, the results of this study validated three metabolites previously reported in other studies, namely 1-arachidonoylglycerophosphocholine, epiandrosterone sulfate, and hexadecanedioate. On the other hand, it establishes the causal relationship between H-BMD and several new lipid metabolites. At the same time, considering the important role of metabolites in metabolic pathways and the crucial role played by metabolic pathways in various biological processes, we obtained information on the metabolites downstream of the obtained lipid metabolites pathway and conducted a causal analysis. Intriguingly, downstream metabolites of the eight lipid metabolites have not shown a causal relationship with H-BMD. This also demonstrates that the causal relationship between those eight lipid metabolites we obtained and H-BMD might be direct, without mediation through the aforementioned metabolites.

Acetylcarnitine and propionylcarnitine are both acyl-carnitine metabolites involved in fatty acid metabolism and energy metabolism pathways [34,35]. Acylcarnitine serves as an important transporter of fatty acids from the cytoplasm to the mitochondria, where it participates in fatty acid oxidation and energy generation [34,36]. Hexadecanedioate and tetradecanedioate are considered biomarkers of transporter function and are metabolic products of ω-oxidation of fatty acids [37,38]. They are involved in the synthesis of long-chain fatty acids in the cytoplasm through the catalysis of a series of enzymes. β-Oxidation breaks down fatty acids into shorter acyl-CoA molecules, and hexadecanedioate and tetradecanedioate are gradually converted into shorter fatty acyl-CoA, ultimately producing metabolites such as pyruvic acid and acetyl-CoA. Acetyl-CoA, the final product of fatty acid metabolism, can enter the citric acid cycle for oxidative metabolism, generating energy and carbon dioxide. Myo-inositol is involved in the inositol metabolism pathway [39] and has been identified as an essential growth-promoting factor for mammalian cells and animals [40]. Sodium/myo-inositol cotransporter-1 is one of the transporters responsible for introducing myo-inositol into cells [41]. Dai et al. has demonstrated the adverse effects of sodium/myo-inositol cotransporter-1 deficiency on prenatal bone development and postpartum bone remodeling, confirming its significant role in BMD [42]. Epiandrosterone sulfate, a decomposition product of dehydroepiandrosterone, plays an essential role in the androgen steroid metabolism pathway [43]. It is commonly used as an endogenous steroid supplement and is often used as a marker for detecting the abuse of stimulants such as testosterone [44,45]. Moayyeri et al. have previously demonstrated the causal relationship between the products of the epiandrosterone sulfate metabolic pathway and hip or spine BMD [46]. 1-linoleoylglycophoethanolamine is an essential component of phosphatidyl ethanolamine (PE) and serves as a biomarker of glycerophospholipid metabolism [47]. PE, the main component of cell membrane phospholipids, plays a crucial role in maintaining cell structure stability [48,49]. It has been reported that PE can promote osteogenesis [50]. Bispo et al. have also shown that the content of PE increases during the osteogenic differentiation of human adipose tissue stem cells [51]. Moreover, previous experimental results by Aleidi et al. have indicated significant dysregulation of PE in patients with low BMD [52]. Additionally, Irie et al. have demonstrated that increased PE content plays a crucial role in osteoclast fusion, and increased osteoclast differentiation and activity are associated with reduced osteoblast differentiation and mineralization [53]. Given the limited and conflicting research results thus far, further studies are needed to evaluate the biological effects and clinical value of 1-linoleoylglycerophoethanolamine and PE on H-BMD. 1-arachidonoylglycerophosphocholine is an important lysophosphatidylcholine [54]. Recently, Jia et al. have identified 1-arachidonoylglycerophophorine as a potential predictive biomarker for psychological disorders [55]. However, there are currently no reports on the association between 1-arachidonoylglycerophosphocholine and H-BMD or BMD.

Age-related variables, genetic factors, changes in hormone levels, unhealthy lifestyle choices, etc. are the main causes of osteoporosis [56]. The current treatment strategy primarily involves medication therapy, examination and monitoring of bone density, and dietary and nutritional improvements [57,58,59]. Medication therapy for osteoporosis has evolved from animal research and clinical observation research to being primarily driven by basic bone biology research [60]. Current areas of study for osteoporosis include etiology, medicine development, preventative methods, and advancements in BMD monitoring equipment [61,62,63]. Continuous research on osteoporosis has led to the discovery of an increasing number of targets of action, among which metabolite is a significant element that cannot be disregarded.

Although several metabonomics studies have reported potential fatty acid metabolites associated with H-BMD, these studies are mostly cross-sectional and suffer from uncontrolled confusion and reverse causal relationships. Therefore, the causal relationship between these lipid metabolites and H-BMD remains unclear. However, the MR method can assess causality without being influenced by confounding factors and reverse causality, providing valuable information for clinical practice. The lipid metabolites which were identified by our study could be several potential biomarkers for osteoporosis or be the target points for nutritional supplementation to improve H-BMD even systemic BMD. However, further functional studies and longitudinal cohort studies are needed to apply the causal relationships in statistics to clinical work.

This study has several strengths. Firstly, we identified several metabolites that have not been previously reported to be causally associated with H-BMD. Secondly, we utilized the known pathway information from KEGG and WikiPathways to explore causal relationships between downstream metabolites and H-BMD. Thirdly, by employing a two-sample MR framework, we minimized observational biases such as confounding factors and reverse causality in metabolomics studies. In summary, our study focused on the causal relationship between lipid metabolites and H-BMD and provided new insights into site-specific related metabolites. However, there are some limitations to consider. The GWAS dataset used in this study only includes European demographics, which can avoid the bias of population stratification. Therefore, because of the delicate genetic heterogeneity, the results may not be directly applicable to other races. Additionally, we did not conduct gender- or age-stratified analysis due to the lack of GWAS data. The small sample size and incomplete inclusion of lipid metabolites as well as the number of SNP variants that have been identified for lipid metabolites in the mGWAS dataset are also limiting factors. Furthermore, the study only investigated the causal relationship between lipid metabolites downstream of the currently known KEGG and WikiPathways and H-BMD, due to the limitations of the KEGG and WikiPathways databases, and did not rule out the role of lipid metabolites through a currently unknown or not included KEGG pathway and WikiPathways. Meanwhile, although DXA is considered the current gold standard for diagnosing osteoporosis, there is currently a lack of data on SNPs of H-BMD determined using DXA. In this manuscript, we obtained our heel BMD data using quantitative ultrasound (QUS). Finally, while low H-BMD can predict fracture risk to some extent, whether the identified candidate metabolites can be translated into clinical practice needs further validation through cohort studies.

## 5. Conclusions

In summary, employing MR analysis within the range of available lipid metabolites, we identified several potential metabolites that exhibit causal effects on H-BMD. Further experimental investigations are warranted to verify these candidate metabolites as biomarkers and to elucidate their underlying mechanisms.

## Figures and Tables

**Figure 1 nutrients-15-04160-f001:**
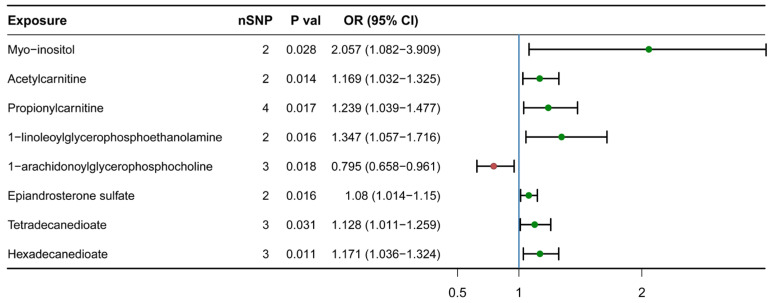
Causal association identified between eight lipid metabolites and H-BMD. The range of OR values for eight lipid metabolites are shown in forest plots. The OR values of each lipid metabolite are shown by the green (OR values > 1) and red (OR values < 1) points, respectively, with the vertical lines on each side of the point denoting the 95% confidence interval. OR, odds ratio; SNP, single-nucleotide polymorphism; 95%CI, 95% confidence interval.

**Figure 2 nutrients-15-04160-f002:**
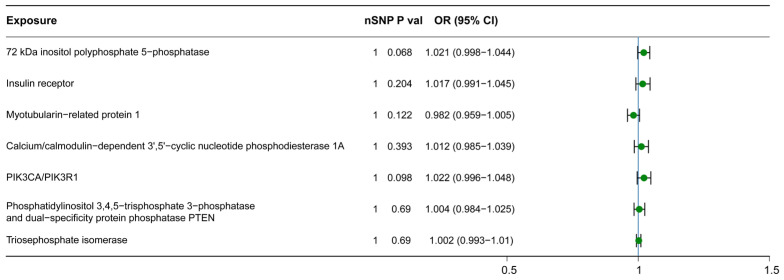
Causal association identified between downstream metabolites and H-BMD. The range of OR values for downstream metabolites are shown in forest plots. The green point (OR values > 1) represents the OR values of each downstream metabolite. The vertical lines on each side of the green point reflect the 95% confidence interval. OR, odds ratio; SNP, single-nucleotide polymorphism; 95%CI, 95% confidence interval.

## Data Availability

The summary-level data of 130 lipid metabolites came from a supporting online website of the study of Shin et al. (http://metabolomips.org/si/, accessed on 21 August 2023). The summary-level data of H-BMD were from UK Biobank https://www.ukbiobank.ac.uk/, accessed on 21 August 2023).

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
