# Peer review of "Mendelian Randomization Study of Lipid Metabolites Reveals Causal Associations with Heel Bone Mineral Density"

_nutrients, 2023, doi:10.3390/nu15194160_

Round 1
Reviewer 1 Report
Comments: The author Zhou et. al in their ms titled “The causal relationship between lipid metabolites and heel 2 bone mineral density: a two-sample Mendelian randomization 3 study” established the hypothesis that lipid metabolites have an impact on bone health. This study has significant implications for the development of new strategies to osteoporosis prevention and treatment. Overall, the ms is quite interesting and well supported by their results and explanation. I have no such major concern about this ms, but till some minor issues need to address to improve the ms before acceptance.
Minor comment:
1. Title need to be change.
2. Author need to be careful about the use of word like ‘ordinary/major’.
3. Line 64: what does it mean by ‘et. al’.
4. Author should address the main cause behind the osteoporosis and what are current treatment option available to conclude their major focus of study.
5. The table and figure used in main ms and supplementary section is very difficult to read.
Author Response
|
Comments 1: Title need to be change.
|
|
Response 1: Thank you for your valuable suggestions. We sincerely appreciate your suggestion regarding the title change. We have reviewed the contents of the manuscript and examined the title of the articles related to Mendelian randomization. Based on your recommendation, we have revised the title to "Mendelian randomization study of lipid metabolites reveals causal associations with heel bone mineral density". We are grateful for your guidance in refining the title, and we believe that the revised title will more accurately and intuitively reflect our research content, thus facilitating readers' comprehension. Thank you once again for your invaluable input and time.
|
|
Comments 2: Author need to be careful about the use of word like ‘ordinary/major’. |
|
Response 2: Agree, thanks for the suggestion. In the Abstract and Introduction sections, we used similar words when introducing osteoporosis. We understand the importance of maintaining precision and clarity in scientific writing, and we are grateful for your diligent review. In consideration of your suggestion, we have removed the relevant words (ordinary/major). [page 1, paragraph 1, and line 16; page 2, paragraph 1, and line 47.] |
|
Comments 3: Line 64: what does it mean by ‘et. al’.
|
|
Response 3: Thank you so much for your careful check. We apologize for any confusion caused by the incorrect use of ‘et. al’ in our paper. Here, it would be more appropriate to utilize ‘etc.’ rather than ‘et. al’. Your suggestion has an improvement effect on the quality of the manuscript. We have made modifications in the manuscript. We feel sorry for the inconvenience brought to the reviewers. [page 2, paragraph 2, and line 63.] |
|
Comments 4: Author should address the main cause behind the osteoporosis and what are current treatment option available to conclude their major focus of study.
|
|
Response 4: Thank you very much for the constructive comments. We greatly appreciate the insights you have provided, as they will significantly enhance the comprehensiveness of our paper. Following a thorough literature search, we have incorporated the following content into the Discussion section: Age-related variables, genetic factors, changes in hormone levels, unhealthy lifestyle choices, etc. are the main causes of osteoporosis [1]. The current treatment strategy primarily involves medication therapy, examination and monitoring of bone density, and dietary and nutritional improvements [2-4]. Medication therapy for osteoporosis has evolved from animal research and clinical observation research to being primarily driven by basic bone biology research [5]. Current areas of study for osteoporosis include etiology, medicine development, preventative methods, and advancements in BMD monitoring equipment [6-8]. Continuous research on osteoporosis has led to the discovery of an increasing number of targets of action, among which metabolite is a significant element that cannot be disregarded. [page 7, paragraph 2, and line 288-297.] |
|
Comments 5: The table and figure used in main ms and supplementary section is very difficult to read. |
|||||||||||||
|
|
|||||||||||||
|
Response 5: Thank you for your careful checks. We sincerely apologize for any confusion and inconvenience caused by the table and figures in our manuscript. We have included two figures that illustrate 8 lipid metabolites (Figure 1) that are causally related to H-BMD, as well as their downstream metabolites (Figure 2). The forest plot is a commonly employed graphical representation in Mendelian randomization studies. In our manuscript, we have made efforts to simplify the figures as much as possible, aiming to facilitate the understanding of both reviewers and readers. In the Supplementary Material, we conducted a search for these statistically significant lipid metabolites in both the KEGG database and the Wikipathys database, and labeled the corresponding target metabolites and their downstream metabolites in the figures. Additionally, we have provided a list of 130 lipid metabolites in Supplementary Table 1, and the downstream metabolites of the aforementioned 8 lipid metabolites in Supplementary Table 2. We have made some adjustments based on your suggestions and have included further details in the figure legends and table legends in the supplementary material. We replace “genes or enzymes” with “molecules” in each supplementary figure legend. Then we added a sentence to each Supplementary Figure legend - "Subsequently, we obtained SNPs for downstream metabolites in the IEU OpenGWAS project and explored their causal relationship with H-BMD. URL of the IEU OpenGWAS project: https://gwas.mrcieu.ac.uk/" We replace “GWAS” with “IEU OpenGWAS project” in the both legend of Supplementary Tables. Then we added a sentence to the legend of Supplementary Table 1: Code: The code corresponding to the metabolites listed in the IEU OpenGWAS project. URL of the IEU OpenGWAS project: https://gwas.mrcieu.ac.uk/. Similarly, we added in the legend of Supplementary Table 2. "EC: Enzyme Commission number in KEGG; GWAS-ID: The corresponding ID of the downstream molecule in the IEU OpenGWAS project. project. URL of the IEU OpenGWAS project: https://gwas.mrcieu.ac.uk/ ".
To provide you with a clearer understanding of the changes, please refer to the following two tables. Table 1: The revisions what we made in the legend of Supplementary Figures.
Table 2: The revisions what we made in the legend of Supplementary Tables.
We would like to thank the referee again for taking the time to review our manuscript. [page 8, paragraph 2, and lines 337-344; Supplemental Material Tables and Supplemental Material Figures] |
- Christiansen, C. Osteoporosis: diagnosis and management today and tomorrow. Bone 1995, 17, 513s-516s, doi:10.1016/8756-3282(95)00345-0.
- Ensrud, K.E.; Crandall, C.J. Osteoporosis. Annals of internal medicine 2017, 167, ITC17-ITC32, doi:10.7326/AITC201708010.
- Taylor, A.; Staruchowicz, L. The Experience and Effectiveness of Nurse Practitioners in Orthopaedic Settings: A Comprehensive Systematic Review. JBI library of systematic reviews 2012, 10, 1-22, doi:10.11124/jbisrir-2012-249.
- Garvey, W.T.; Mechanick, J.I.; Brett, E.M.; Garber, A.J.; Hurley, D.L.; Jastreboff, A.M.; Nadolsky, K.; Pessah-Pollack, R.; Plodkowski, R. AMERICAN ASSOCIATION OF CLINICAL ENDOCRINOLOGISTS AND AMERICAN COLLEGE OF ENDOCRINOLOGY COMPREHENSIVE CLINICAL PRACTICE GUIDELINES FOR MEDICAL CARE OF PATIENTS WITH OBESITY. Endocrine practice : official journal of the American College of Endocrinology and the American Association of Clinical Endocrinologists 2016, 22 Suppl 3, 1-203, doi:10.4158/ep161365.Gl.
- Khosla, S.; Hofbauer, L.C. Osteoporosis treatment: recent developments and ongoing challenges. Lancet Diabetes Endocrinol 2017, 5, 898-907, doi:10.1016/S2213-8587(17)30188-2.
- Zhou, S.; Tao, Z.; Zhu, Y.; Tao, L. Mapping theme trends and recognizing hot spots in postmenopausal osteoporosis research: a bibliometric analysis. PeerJ 2019, 7, e8145, doi:10.7717/peerj.8145.
- Mundy, G.R. Nutritional modulators of bone remodeling during aging. The American journal of clinical nutrition 2006, 83, 427s-430s, doi:10.1093/ajcn/83.2.427S.
- Tosi, L.L.; Kyle, R.F. Fragility fractures: the fall and decline of bone health. Commentary on "Interventions to improve osteoporosis treatment following hip fracture" by Gardner et Al. The Journal of bone and joint surgery. American volume 2005, 87, 1-2, doi:10.2106/jbjs.D.02881.

Reviewer 2 Report
Dear Authors,
Your publication presents a comprehensive and robust analysis of the causal relationship between lipid metabolites and heel bone mineral density, providing valuable insights into the potential role of lipid metabolism in bone health. The use of two-sample Mendelian randomization analysis is a strength of this study, as it is a powerful tool for identifying causal relationships in observational data. The identification of specific metabolites that have a causal relationship with heel bone mineral density could have significant implications for the development of new strategies for osteoporosis prevention and treatment. The study also includes a metabolomic pathway analysis, which provides valuable insights into the potential mechanisms underlying the observed associations. Overall, your study makes a valuable contribution to our understanding of the role of lipid metabolism in bone health and opens up new avenues for future research. It is an interesting publication that brings much to the search for pharmacological points of grip in both prevention and therapy of osteoporosis.
Author Response
|
Comments 1: Dear Authors, Your publication presents a comprehensive and robust analysis of the causal relationship between lipid metabolites and heel bone mineral density, providing valuable insights into the potential role of lipid metabolism in bone health. The use of two-sample Mendelian randomization analysis is a strength of this study, as it is a powerful tool for identifying causal relationships in observational data. The identification of specific metabolites that have a causal relationship with heel bone mineral density could have significant implications for the development of new strategies for osteoporosis prevention and treatment. The study also includes a metabolomic pathway analysis, which provides valuable insights into the potential mechanisms underlying the observed associations. Overall, your study makes a valuable contribution to our understanding of the role of lipid metabolism in bone health and opens up new avenues for future research. It is an interesting publication that brings much to the search for pharmacological points of grip in both prevention and therapy of osteoporosis. |
|
Response 1: Thank you for your encouraging comment! We appreciate your thorough evaluation of our study, and we are pleased to hear that you found our analysis to be comprehensive and robust. We agree that the use of two-sample Mendelian randomization analysis is a powerful tool for identifying causal relationships in observational data. By utilizing this approach, we were able to identify specific metabolites with a causal relationship to heel bone mineral density, which has important implications for the development of new strategies for osteoporosis prevention and treatment. We are glad that you found our study interesting and valuable in the search for pharmacological points of grip in both the prevention and therapy of osteoporosis. We hope that our findings will spur further investigation into the relationship between lipid metabolism and bone health and ultimately lead to improved clinical outcomes for patients. Thank you again for your thoughtful review, and we look forward to addressing any additional feedback you may have. |
Reviewer 3 Report
The submitted study adds some new information to the basic knowledge on the relationship between lipid metabolites and heel bone mineral density however, I have several major comments.
1. For better understanding the Methods section should be complemented with more precise informations on the methods used for measurement of lipid metabolites and whether heel BMD was measured by QUS or other method. QUS can be applied as a prescreening tool in osteoporosis management but according to recent data is not associated with future fractures which makes it useless for patient monitoring. Currently, dual-energy x-ray absorptiometry (DEXA) is the gold standard for diagnosis of osteoporosis and monitoring osteoporosis treatment.
2. What is of importance, the authors should also comment on practical implications of their findings as the use of metabolomics for measurement of lipid metabolites is absolutely not widely available and limited to specialized research laboratories. This issue should be much more widely explained.
Author Response
|
Comments 1: The submitted study adds some new information to the basic knowledge on the relationship between lipid metabolites and heel bone mineral density however, I have several major comments. 1.For better understanding the Methods section should be complemented with more precise informations on the methods used for measurement of lipid metabolites and whether heel BMD was measured by QUS or other method. QUS can be applied as a prescreening tool in osteoporosis management but according to recent data is not associated with future fractures which makes it useless for patient monitoring. Currently, dual-energy x-ray absorptiometry (DEXA) is the gold standard for diagnosis of osteoporosis and monitoring osteoporosis treatment.
|
|
Response 1: Special thanks to you for your heuristic comment. We fully agree with your comment and would like to provide further elucidation on this matter. The data used in our study is derived from two publicly available datasets, namely the information on lipid metabolites in the blood and Heel Bone Mineral Density (H-BMD) [1,2]. The data on lipid metabolites was extracted from a genome-wide association study (GWAS) dataset published by Shin et al. in 2014 [1]. This study conducted a comprehensive analysis of the impact of genetics on human metabolism through genome-wide association analysis of metabolites in blood samples. The H-BMD data, on the other hand, was obtained from a large-scale GWAS meta-analysis conducted by Morris et al. in 2019 [2]. In our study, H-BMD was measured using quantitative ultrasound (QUS) of the heel. While dual-energy X-ray absorptiometry (DXA) is considered the gold standard for diagnosing and treating osteoporosis, the use of QUS is also widespread. Previous reports have often used heel QUS as a screening tool for predicting osteoporosis [3,4]. QUS has been studied for its correlation with fracture occurrence and its relationship with DXA [5-9]. QUS measurements of the heel and bone mineral density have been shown to be significant predictors of incident vertebral fractures. The relative risks for QUS measurements at the heel are similar to those for DXA measurements[6]. A study conducted in Taiwan also reported a significant correlation between QUS and DXA among the Taiwanese population [7]. According to a study conducted in 2016, there is a significant association between decreased QUS values and an elevated risk of fractures, particularly hip fractures[8]. Furthermore, the accuracy of QUS and DXA in predicting fracture rates in elderly men and women is similar [9]. Therefore, according to Morris et al.'s research, although our data on H-BMD comes from QUS, it is still reliable [2]. However, we were unable to explore the causal relationship between lipid metabolites and H-BMD measured by DXA, as there is no available SNP data for H-BMD based on DXA assays. We have acknowledged this limitation in the Discussion section of our manuscript. If SNP data for H-BMD measured by DXA are provided in future studies, we plan to utilize this data to further investigate the causal relationship between lipid metabolites and H-BMD measured by DXA. The following has been added to the manuscript. “By conducting genome-wide association studies of metabolites in blood samples, Shin and his colleagues give an extensive study of the influences of genes on human metabolism.” [page 3, paragraph 2, lines 106-108] “In addition, quantitative ultrasound (QUS) of the heel was used to obtain the H-BMD data.” [page 3, paragraph 2, lines 112-113] “Meanwhile, although DXA is considered the current gold standard for diagnosing osteoporosis, there is currently a lack of data on SNPs of H-BMD determined using DXA. In this manuscript, we obtained our heel BMD data using QUS.” [page 7, paragraph 4, lines 326-329]
|
|
Comments 2: 2.What is of importance, the authors should also comment on practical implications of their findings as the use of metabolomics for measurement of lipid metabolites is absolutely not widely available and limited to specialized research laboratories. This issue should be much more widely explained. |
|
Response 2: Your insightful suggestions are greatly appreciated. The direct involvement of many endogenous small molecule compounds detected by metabolomics in various metabolic/circular processes within the body, their levels to some extent reflect the function and status of biochemical metabolism. Metabolite-based analysis allows us to explore the relationship between biochemical metabolism and diseases, and investigate and reveal the etiology and pathological mechanisms of diseases from the perspective of related metabolic abnormalities. It also contributes to the discovery of new drug targets. Therefore, we believe that metabolomics research extends beyond the laboratory and holds tremendous potential for wide-ranging applications in clinical settings. While there remains a disparity between our current findings and their direct clinical applications, this study offers valuable evidence and potential research for further exploration of the intricate relationship between osteoporosis and bone density. On this point, we mention it in the Discussion section when we describe the limitations of this manuscript. With the advancement of technology and the reduction in the cost of metabolomics, we firmly believe that metabolite determination through metabolomics may become widely utilized in clinical applications in the future. For instance, Brindle et al. demonstrated a metabolomics-based diagnostic technique for coronary heart disease that is accurate, rapid, and noninvasive[10]. Marchesi et al. also used metabolomics to provide a novel, non-invasive diagnostic method for gastrointestinal diseases[11]. As a matter of fact, several patented methods for clinical disease diagnosis using metabolomics methods have emerged, providing a new approach for some clinical diseases that lack clear diagnostic indexes, which will surely be beneficial to the wide application of metabolomics in disease diagnosis[12,13]. The 8 lipid metabolites we identified as causally related to H-BMD could potentially serve as biomarkers for early detection, prevention, and treatment of osteoporosis patients. Once again, we sincerely thank you for your valuable comment. |
- Shin, S.Y.; Fauman, E.B.; Petersen, A.K.; Krumsiek, J.; Santos, R.; Huang, J.; Arnold, M.; Erte, I.; Forgetta, V.; Yang, T.P., et al. An atlas of genetic influences on human blood metabolites. Nature genetics 2014, 46, 543-550, doi:10.1038/ng.2982.
- Morris, J.A.; Kemp, J.P.; Youlten, S.E.; Laurent, L.; Logan, J.G.; Chai, R.C.; Vulpescu, N.A.; Forgetta, V.; Kleinman, A.; Mohanty, S.T., et al. An atlas of genetic influences on osteoporosis in humans and mice. Nature genetics 2019, 51, 258-266, doi:10.1038/s41588-018-0302-x.
- Metrailler, A.; Hans, D.; Lamy, O.; Gonzalez Rodriguez, E.; Shevroja, E. Heel quantitative ultrasound (QUS) predicts incident fractures independently of trabecular bone score (TBS), bone mineral density (BMD), and FRAX: the OsteoLaus Study. Osteoporos Int 2023, 34, 1401-1409, doi:10.1007/s00198-023-06728-4.
- Thomsen, K.; Jepsen, D.B.; Matzen, L.; Hermann, A.P.; Masud, T.; Ryg, J. Is calcaneal quantitative ultrasound useful as a prescreen stratification tool for osteoporosis? Osteoporos Int 2015, 26, 1459-1475, doi:10.1007/s00198-014-3012-y.
- Yang, K.C.; Wang, S.T.; Lee, J.J.; Fann, J.C.; Chiu, S.Y.; Chen, S.L.; Yen, A.M.; Chen, H.H.; Chen, M.K.; Hung, H.F. Bone mineral density as a dose-response predictor for osteoporosis: a propensity score analysis of longitudinal incident study (KCIS no. 39). QJM 2019, 112, 327-333, doi:10.1093/qjmed/hcz009.
- Hollaender, R.; Hartl, F.; Krieg, M.A.; Tyndall, A.; Geuckel, C.; Buitrago-Tellez, C.; Manghani, M.; Kraenzlin, M.; Theiler, R.; Hans, D. Prospective evaluation of risk of vertebral fractures using quantitative ultrasound measurements and bone mineral density in a population-based sample of postmenopausal women: results of the Basel Osteoporosis Study. Ann Rheum Dis 2009, 68, 391-396, doi:10.1136/ard.2007.083618.
- Yen, C.C.; Lin, W.C.; Wang, T.H.; Chen, G.F.; Chou, D.Y.; Lin, D.M.; Lin, S.Y.; Chan, M.H.; Wu, J.M.; Tseng, C.D., et al. Pre-screening for osteoporosis with calcaneus quantitative ultrasound and dual-energy X-ray absorptiometry bone density. Sci Rep 2021, 11, 15709, doi:10.1038/s41598-021-95261-7.
- Esmaeilzadeh, S.; Cesme, F.; Oral, A.; Yaliman, A.; Sindel, D. The utility of dual-energy X-ray absorptiometry, calcaneal quantitative ultrasound, and fracture risk indices (FRAX® and Osteoporosis Risk Assessment Instrument) for the identification of women with distal forearm or hip fractures: A pilot study. Endocrine research 2016, 41, 248-260, doi:10.3109/07435800.2015.1120744.
- Moayyeri, A.; Adams, J.E.; Adler, R.A.; Krieg, M.A.; Hans, D.; Compston, J.; Lewiecki, E.M. Quantitative ultrasound of the heel and fracture risk assessment: an updated meta-analysis. Osteoporos Int 2012, 23, 143-153, doi:10.1007/s00198-011-1817-5.
- Brindle, J.T.; Antti, H.; Holmes, E.; Tranter, G.; Nicholson, J.K.; Bethell, H.W.L.; Clarke, S.; Schofield, P.M.; McKilligin, E.; Mosedale, D.E., et al. Rapid and noninvasive diagnosis of the presence and severity of coronary heart disease using 1H-NMR-based metabonomics. Nature medicine 2002, 8, 1439-1445, doi:10.1038/nm1202-802.
- Marchesi, J.R.; Holmes, E.; Khan, F.; Kochhar, S.; Scanlan, P.; Shanahan, F.; Wilson, I.D.; Wang, Y. Rapid and noninvasive metabonomic characterization of inflammatory bowel disease. Journal of proteome research 2007, 6, 546-551, doi:10.1021/pr060470d.
- Kaddurah-Daouk, R.; Kristal Bruce, S. METHODS FOR DRUG DISCOVERY, DISEASE TREATMENT, AND DIAGNOSIS USING METABOLOMICS. 2008/07/30/Application date, 2008.
- Laaksonen, R.; Oresic, M.; Lehtimaeki, T.; Paeivae, H. DIAGNOSTIC METHOD FOR MYOPATHY. 2007/06/12/Application date, 2007.

Round 2
Reviewer 3 Report
The manuscript has been sufficiently improved; revised according to my suggestions.
